# Spatially resolved transcriptomics reveals pro-inflammatory fibroblast involved in lymphocyte recruitment through CXCL8 and CXCL10

Ana J Caetano[1†], Yushi Redhead[1], Farah Karim[1,2], Pawan Dhami[3], Shichina Kannambath[3], Rosamond Nuamah[3], Ana A Volponi[1], Luigi Nibali[4], Veronica Booth[4], Eleanor M D'Agostino[5], Paul T Sharpe[1,6*]

[1]Centre for Craniofacial and Regenerative Biology, Faculty of Dentistry, Oral & Craniofacial Sciences, King's College London, London, United Kingdom; [2]Department of Endodontics, Faculty of Dentistry, Oral & Craniofacial Sciences, King's College London, London, United Kingdom; [3]NIHR BRC Genomics Research Platform, Guy's and St Thomas' NHS Foundation Trust, King's College London School of Medicine, Guy's Hospital, London, United Kingdom; [4]Department of Periodontology, Faculty of Dentistry, Oral & Craniofacial Sciences, King's College London, London, United Kingdom; [5]Unilever R&D, Colworth Science Park, Sharnbrook, United Kingdom; [6]Laboratory of Odontogenesis and Osteogenesis, Institute of Animal Physiology and Genetics, Brno, Czech Republic

*For correspondence:
paul.sharpe@kcl.ac.uk

Present address: †Centre for Oral Immunobiology and Regenerative Medicine, Barts and The London School of Medicine and Dentistry, Queen Mary University of London, London, United Kingdom

**Abstract** The interplay among different cells in a tissue is essential for maintaining homeostasis. Although disease states have been traditionally attributed to individual cell types, increasing evidence and new therapeutic options have demonstrated the primary role of multicellular functions to understand health and disease, opening new avenues to understand pathogenesis and develop new treatment strategies. We recently described the cellular composition and dynamics of the human oral mucosa; however, the spatial arrangement of cells is needed to better understand a morphologically complex tissue. Here, we link single-cell RNA sequencing, spatial transcriptomics, and high-resolution multiplex fluorescence *in situ* hybridisation to characterise human oral mucosa in health and oral chronic inflammatory disease. We deconvolved expression for resolution enhancement of spatial transcriptomic data and defined highly specialised epithelial and stromal compartments describing location-specific immune programs. Furthermore, we spatially mapped a rare pathogenic fibroblast population localised in a highly immunogenic region, responsible for lymphocyte recruitment through *CXCL8* and *CXCL10* and with a possible role in pathological angiogenesis through *ALOX5AP*. Collectively, our study provides a comprehensive reference for the study of oral chronic disease pathogenesis.

## Editor's evaluation

The work by Caetano et al. describes the changes caused by periodontal inflammation in terms of tissue structure through integrating multi-omics techniques and fluorescence in situ hybridization. They defined highly specialized epithelial and stromal compartments and spatially mapped a rare pathogenic fibroblast population likely responsible for lymphocyte recruitment and angiogenesis. They also compared the genes with altered expression in gingival inflammation with the ones highlighted from the GWAS analysis related to patients with periodontitis which contributes to generating new hypotheses for future studies.

## Introduction

The human oral mucosa separates and protects deeper tissues and organs in the oral region from the external environment. The main tissue components are a stratified squamous epithelium and an underlying connective tissue layer, called the *lamina propria,* and both compartments undergo dynamic changes in response to disease. Oral mucosa also shows significant structural variation in distinct regions of the oral cavity, and three main types are recognised according to their primary function: masticatory mucosa, lining mucosa, and specialised mucosa. Despite its similarities to human skin, oral mucosa undergoes faster cellular turnover and shows limited scarring following injury (*Iglesias-Bartolome et al., 2018*).

Masticatory mucosa or *gingiva* (tooth-associated mucosa) is the first structure to be affected by an oral chronic disease – periodontitis, which represents the 11th more prevalent worldwide disease and is highly correlated with the worsening of systemic conditions, such as diabetes and cardiovascular disease (*Kinane et al., 2017*). Recent studies from our lab and others have described the oral mucosa transcriptional profile demonstrating that it is a highly heterogeneous and complex tissue with dynamic recruitment and expansion and/or depletion of various cell types (*Caetano et al., 2021*; *Huang et al., 2021*; *Williams et al., 2021*); however, understanding their joint interaction and consequently their functionality *in situ* is not well understood. The isolation and characterisation of single cells at a high resolution is valuable, but it is still insufficient for understanding complex biological behaviours. These analyses still approach cells as isolated functional units, and while it may be reasonable that a response is determined by a single cell type in a particular state, an alternative hypothesis is that the interaction of multiple cells in diverse states is what drives response to disease or therapy.

Recent years have witnessed the advances in spatial transcriptomic and computational technologies. These tools enable untargeted and comprehensive capture of cellular transcriptomic profiles *in situ*, and therefore, can overcome limitations of *in situ* hybridisation (ISH), and allow discoveries of previously undetectable transcriptomic changes. The cellular architecture of tissues, where distinct cell types are organised in space, underlies cell–cell communication, organ function, and pathology as physical proximity often reflects relatedness. Thus, the spatial information of cell transcriptomes provides information about how cells coordinate to perform their biological functions in cell–cell signalling mechanisms.

In this study, we aimed to understand how oral mucosa tissue organisation (structure) is related to tissue physiology (function) by applying unbiased spatial transcriptomics methods and high-resolution microscopy. By studying spatial cellular interactions, we aimed to explain the variation observed within cell types in dissociation-based protocols. We characterise a rare immunogenic fibroblast population, which supports our and other studies on the emerging role of stromal cells in shaping tissue pathology by controlling local immune responses (*Croft et al., 2019*; *Davidson et al., 2021*; *Kinchen et al., 2018*). Understanding the spatial tissue context determining fibroblast–immune cell interactions is vital to identify location-specific signalling events shaping disease, whilst offering novel therapeutic avenues to patients with chronic infection and inflammation. In short, we exploit high-throughput spatial transcriptomics and scRNA-seq to create a large-scale tissue atlas of human oral mucosa in health and disease, charting changes across location and tissue compartments.

## Results

### Molecular spatial reconstruction of the human oral mucosa in health

The function of many biological systems depends on the spatial organisation of their cells; genes need to be properly regulated in space for a system to function. To systematically study the spatial patterns of gene expression of human oral mucosa that we previously described (*Caetano et al., 2021*) and understand *in situ* tissue organisation, healthy human oral mucosa samples were analysed by spatial transcriptomics (ST) (10X Genomics Visium) (*Figure 1A*). Two distinct anatomical locations were used in our analyses – buccal (anterior; *Figure 1B–F*) and palatal (posterior; *Figure 1—figure supplement 1*) from three individuals. In each experiment, 10-µm-thick paraffin and 15-µm-thick cryosection slices from five individual healthy samples in total were analysed. We generated two technical replicates for two biological replicates to understand technical variability and no major differences were detected (*Figure 1—figure supplements 1 and 2*). Sequencing data were processed and integrated to generate spatial transcriptomes of each section. Typically, this captured more than 40,000

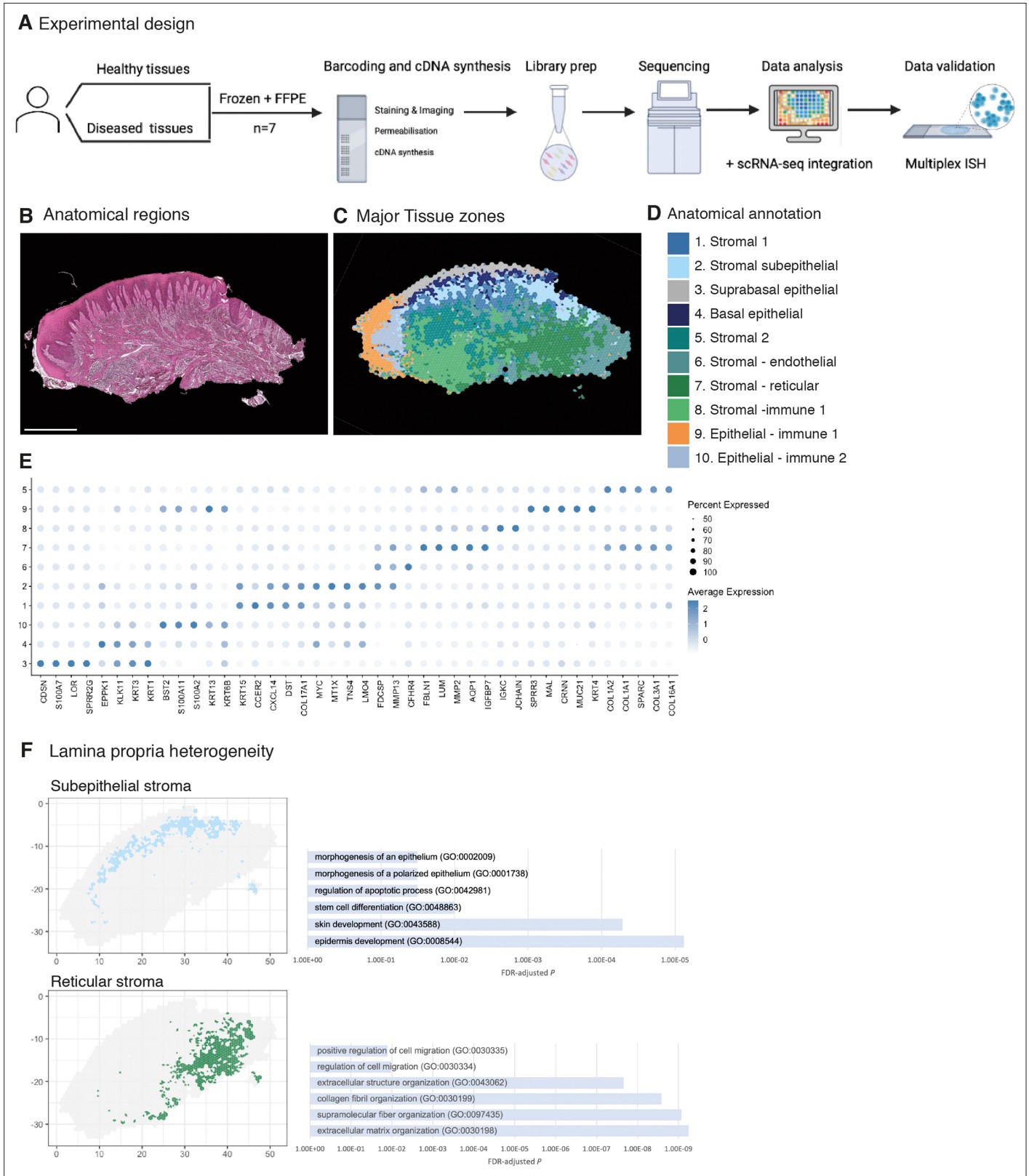

**Figure 1.** Spatial map of the human oral mucosa. (**A**) Overview of study design for human oral mucosa. A total of nine tissue sections from seven distinct patients were sequenced and analysed. (**B**) Haematoxylin and eosin (H&E) image of a representative healthy oral mucosa section demonstrating general tissue morphology and demarcation between the two major tissue compartments, epithelium and connective tissue. Scale bar: 1 mm. (**C**) Human oral mucosa regions present in the assayed section. (**D**) Anatomical annotation of unbiased transcriptional tissue regions. (**E**) Markers of tissue compartment-

*Figure 1 continued on next page*

*Figure 1 continued*

specific genes used for tissue annotation showing percent of expressing cells (circle size) and average expression (colour) of gene markers (rows) across compartments (columns). (**F**) Connective tissue (lamina propria) heterogeneity Gene Ontology (GO) analyses showing subepithelial region enrichment for epithelium development and deep reticular region enriched for extracellular matrix (ECM) terms.

The online version of this article includes the following source data and figure supplement(s) for figure 1:

**Source data 1.** Top spatially variable features in health (referent to *Figure 1*).

**Figure supplement 1.** Mapping of palatal human oral mucosa and top differentially expressed genes.

**Figure supplement 2.** Biological and technical replicates used for ST analyses.

mean reads under tissue per spot representing over 2000 genes per 55 µm spot. We then performed unsupervised clustering, and clusters were associated with tissue structure, followed by manual validation and annotation.

Analyses of transcriptional signatures of ST spots identified between 4 and 10 oral mucosa compartments in each slide, which mapped to distinct locations and allowed us to define anatomical regions from a molecular perspective (*Figure 1C–E*, *Figure 1—figure supplements 1 and 2*, *Figure 1—source data 1*, *Figure 2—source data 1*).

Since ST spots measure 55 µm, these contain multiple cell types. To address this low resolution, we used a new computational tool to deconvolute each spot and predict the underlying composition of cell types (*Zhao et al., 2021*). This method uses neighbour structure in spatial transcriptomic data to increase the resolution at a subspot level and infers from the existing literature for use of Bayesian statistics to achieve high-resolution images. Thus, we were able to improve our mapping that allowed us to predict gene expression at higher resolution (*Figure 1D*). Transcriptional regions were annotated based on known epithelial and stromal transcriptional signatures and histological assessment (*Figure 1C–E*, *Figure 1—figure supplement 1*, *Figure 1—source data 1*). Most cluster regions were distributed in layers, highlighting that the largest determinant of transcriptional/cellular spatial variability corresponds to tissue depth as described in other barrier organs (*Fawkner-Corbett et al., 2021*). We further investigated each individual region and generated spatially variable features that we used to annotate each region. We found molecular patterns that mapped into known anatomical areas; for example, *KRT1* mapped into the suprabasal epithelial region, and *KRT15* into the basal epithelial layer. Within the less characterised lamina propria we observed transcriptionally distinct regions with possibly distinct cellular functions; for example, the subepithelial stromal region shows specific GO terms for epidermis development, stem cell differentiation, and morphogenesis of the epithelium, which follows recent research on subepithelial niche areas (*Kinchen et al., 2018*; *Shoshkes-Carmel et al., 2018*). By contrast, lower reticular regions show mostly ECM organisation terms (*Figure 1F*). These results follow the known compartmentalised of papillary and reticular fibroblasts in the skin (*Plikus et al., 2021*).

Altogether, our analyses yield the first comprehensive molecular *in situ* description of the human oral mucosa in health, with the discovery of transcriptionally distinct regions.

## Remodelling of the human oral mucosa structure in oral chronic inflammatory disease

To understand how tissue organisation is reshaped in disease, we investigated changes in tissue organisation and gene expression in patients with periodontal disease. We again undertook ST (10X Visium) on tissue from four different patient samples. Analyses of transcriptional signatures of ST spots identified between 5 and 10 spot clusters in each slide, which again mapped to discreet anatomical locations (*Figure 2A–E*). We readily identified similar cluster regions as detected in healthy mucosa, and we observed hallmarks of a pro-inflammatory tissue microenvironment in disease (*Figure 2E*, *Figure 1—figure supplement 1*, *Figure 2—source data 1*). To validate the spatial transcriptomes that we obtained, we compared spatial expression patterns of top marker genes of oral mucosa tissue with known tissue markers. The spatial patterns of these transcripts matched well with those of immunohistochemistry (IHC) results (*Figure 2F*). We also observed several disease-associated changes in the stromal compartment, exemplified by an overall upregulation of pro-inflammatory gene signatures (*Figure 2E*, *Figure 1—figure supplement 1*, *Figure 2—source data 1*), and upregulation of genes involved in vascular transport (*CXCL12*) and blood vessel morphogenesis (*CLDN5*,

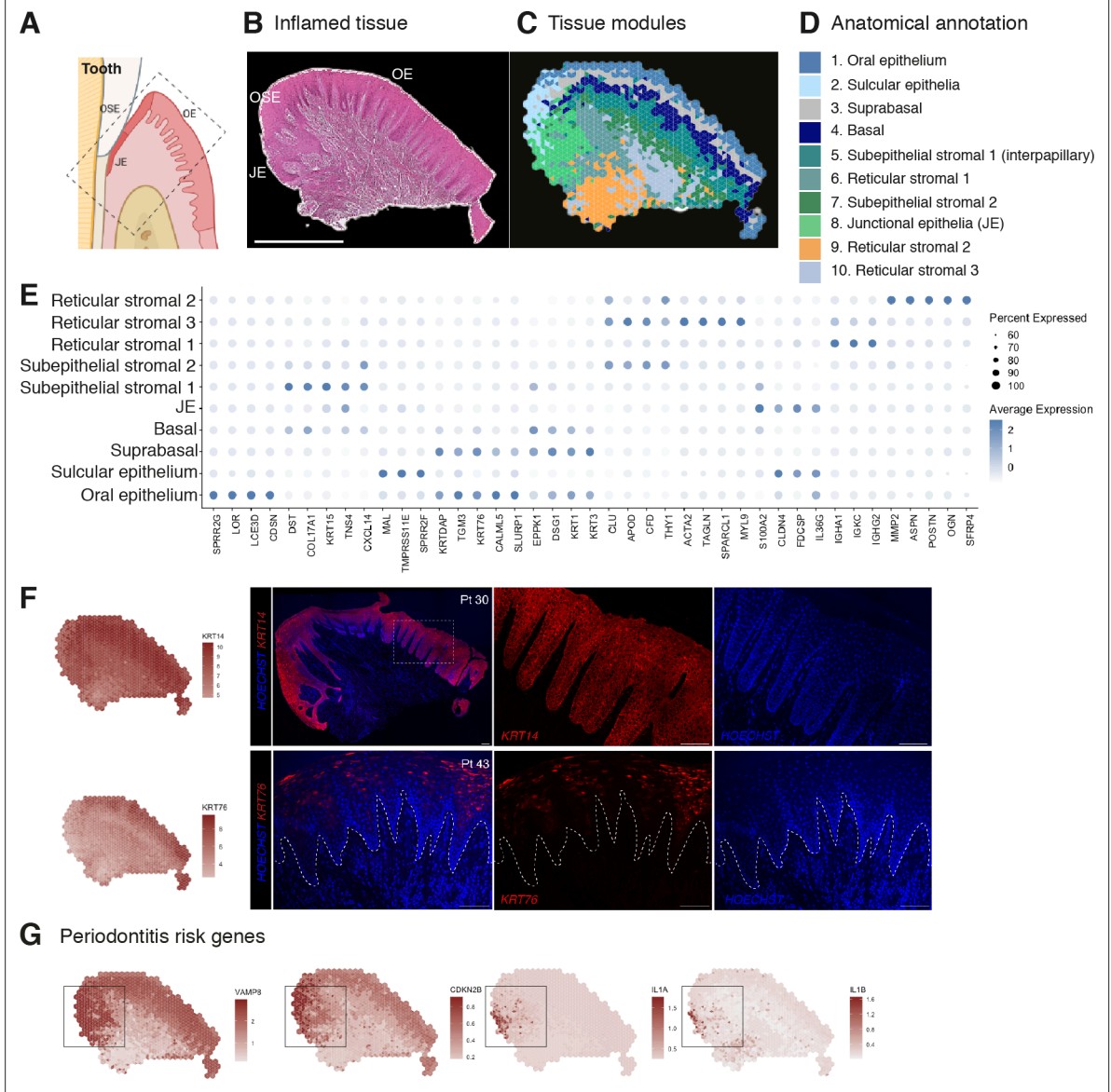

**Figure 2.** Human oral mucosa structure in oral chronic inflammatory disease. (**A**) Schematic illustration of the human oral mucosa showing the different epithelial regions, oral epithelium (OE), oral sulcular epithelium (OSE), and junctional epithelium (JE). Created with Biorender. (**B**) H&E image of the representative inflamed oral mucosa section demonstrating demarcation between the distinct epithelia, and connective tissue region. (**C**) Human oral mucosa regions present in the assayed section using BayesSpace. (**D**) Anatomical annotation of unbiased transcriptional tissue regions. (**E**) Markers of tissue compartment differentially expressed genes used for tissue annotation showing percent of expressing cells (circle size) and average expression (colour) of gene markers (rows) across compartments (columns). (**F**) Immunofluorescence image validation stained for *KRT14* and *KRT76* (representative image, n = 3 samples). Scale bars: 100 μm. (**G**) Mapping of periodontitis risk genes showing *VAMP8, CDKN2B, IL1A,* and *IL1B* restricted expression in the junctional epithelium region.

The online version of this article includes the following source data for figure 2:

**Source data 1.** Top spatially variable features in disease (referent to *Figure 2*).

*TAGLN*) (*Figure 2E*). These results support the hypothesis that tissue-specific properties reflect local organisation.

To reveal transcriptional defects that might drive disease pathogenesis, we investigated spatial gene expression of genes known to be linked to periodontitis susceptibility from recent GWAS that revealed single-nucleotide polymorphisms (*Caetano et al., 2022*; *Nibali et al., 2019*). Mapping of 21 known disease genes with our spatial data allowed us to link genotypes with phenotypes that likely manifest through highly cell-type and location-specific defects and result in pathology. From these,

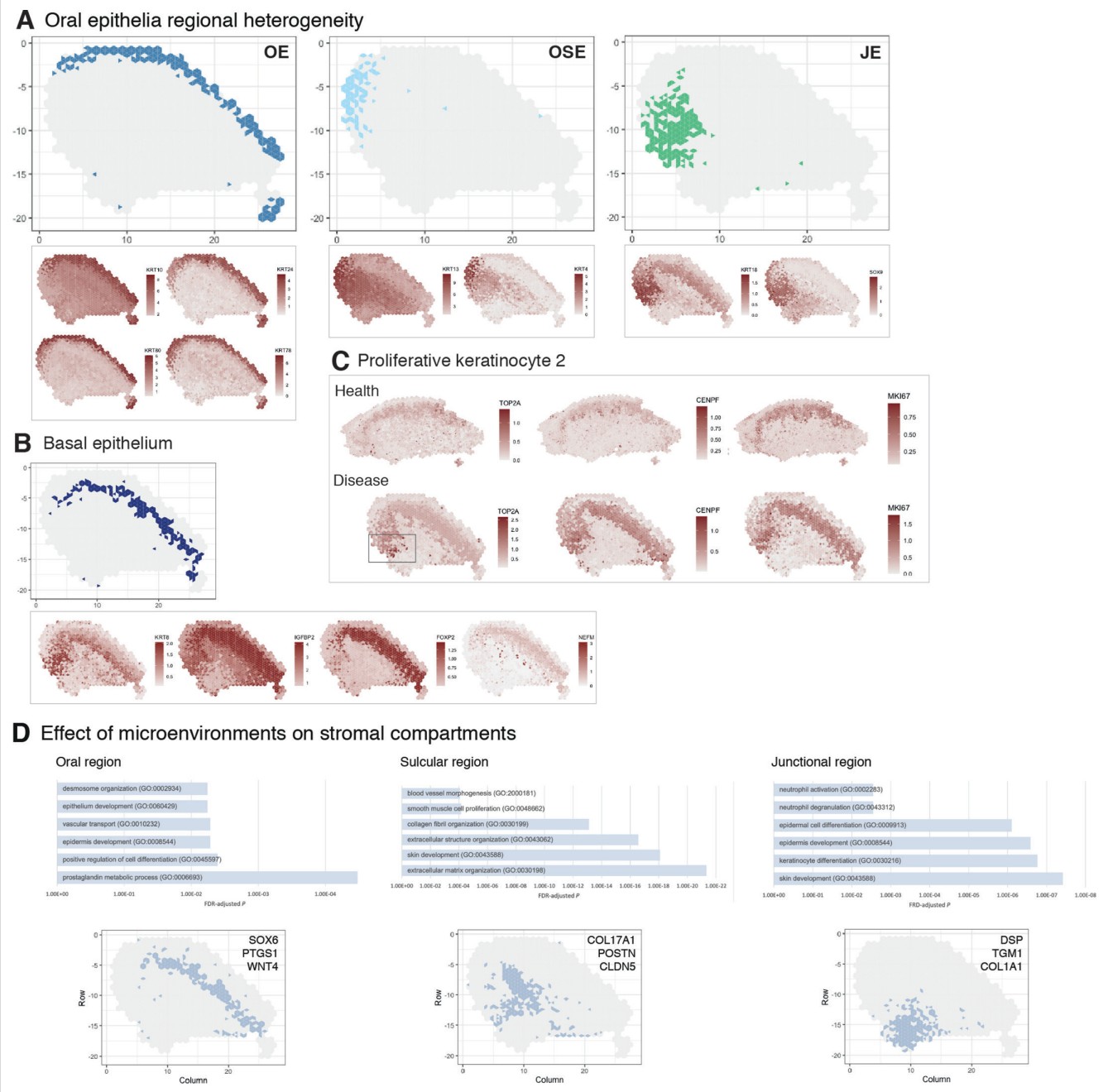

**Figure 3.** Characterisation of the human oral mucosa epithelium. (**A**) Spatial feature plots showing expression of distinct keratins that define each oral epithelia region. (**B**) Characterisation of the proliferative basal epithelial layer showing top differentially expressed genes, *KRT8, IGFBP2, FOXP2, NEFM.* (**C**) Mapping of a rare proliferative epithelial population increased in oral chronic inflammatory disease defined by expression of *TOP2A* and *CENPF. MIKI67* expression showing increased expression in disease (representative images, n = 7). (**D**) The top GO terms associated with the differentially expressed genes in three distinct stromal regions associated with the three distinct oral epithelia; FDR, false discovery rate.

*CDKN2B, IL1A, IL1B,* and *VAMP8* specifically localised in the junctional epithelium region (*Figure 2G*). In summary, our analysis suggests that dysfunctional junctional epithelium might be a major driver of periodontal disease pathogenesis.

## Human oral epithelia regional heterogeneity

The most fundamental function of an epithelial tissue is to act as a barrier; the gingival epithelium can be divided into three distinct anatomical regions – oral (OE), sulcular (OSE), and junctional epithelium

(JE) (*Figure 3A*). In health, the JE attaches to the tooth, providing a seal against oral microorganisms, thus having a profound role in the innate response. Breakdown of this attachment results in loss of periodontal ligament, alveolar bone resorption, and often, tooth loss. Despite recent evidence of epithelial proliferative and stem cell heterogeneity in mouse models (*Byrd et al., 2019*; *Tanaka et al., 2021*; *Yuan et al., 2021*), to date it has not been possible to distinguish between these regions at a transcriptional level.

Since we were able to collect intact tissues that incorporated the entire gingival margin, our analyses were able to unbiasedly distinguish between all three types of epithelia based on their gene expression (*Figure 3A*), suggesting that these are transcriptionally distinct.

Oral epithelium is enriched in genes related to integrin-mediated signalling pathway and keratinocyte differentiation, whereas both sulcular and junctional epithelia show mostly terms related to immune response (*Figure 2E*). They also show distinct keratin signatures with *KRT13* and *KRT4* being restricted to the sulcular epithelium and *KRT18* specific to the junctional epithelium (*Figure 3A*). Interestingly, *SOX9* shows a restricted expression to the junctional epithelium, and it is known to be expressed in stem cells/progenitors in several tissues such as gut and skin (*Jo et al., 2014*), and may label similar populations in the human oral mucosa (*Figure 3A*).

We also analysed the basal proliferative epithelial progenitor layer, which shows high expression of *KRT8, IGFBP2, FOXP2,* and *NEFM* (neurofilament medium chain) (*Figure 3B*). Furthermore, in our reference dataset we identified a rare disease-associated epithelial proliferative population characterised by *TOP2A* and *CENPF* expression. When mapped into the spatial data, we observed that expression of *TOP2A* was highly restricted to the junctional epithelium and significantly increased when compared to healthy samples. Other proliferation markers were also increased in disease (*Figure 3C*). Thus, an inflammatory environment leads to an increased basal epithelial proliferation, suggesting that chronic inflammatory disease may regulate stem cell behaviour.

Furthermore, distinct stromal regions also clustered adjacent to the different epithelial regions similar to healthy samples, suggestive of playing a part in epithelial specification or dysregulation (*Figure 3D*). Within the junctional epithelium area, we observed a distinct pro-inflammatory stromal signature with specific expression of genes involved in 'neutrophil activation' and 'neutrophil degranulation' (*Figure 3D*; *Figure 1—figure supplement 1*); whereas, the stromal sulcular epithelium region showed genes involved in angiogenesis (*Figure 3D*, *Figure 1—figure supplement 1*).

## Stromal–immune–endothelial interactions characterise diseased tissue microenvironment

Cell–cell interactions are the basis of organ development, homeostasis, and disease (*Bonnans et al., 2014*; *Rouault and Hakim, 2012*; *Tucker and Sharpe, 2004*). To investigate how surrounding cells may shape signalling in the oral mucosa across health and disease, we used a cell–cell communication pipeline that considers spatial cellular colocalisation when mapping ligand–receptor pairs (*Dries et al., 2021*). This method can identify which ligand–receptor pairs are potentially more or less active when cells from two cell types are spatially adjacent to each other. Thus, we generated a spatial neighbourhood network (*Figure 4A*), where in health we observed dominant interactions between suprabasal and basal epithelial layers (*MMP9-LRP1; ADAM17-NOTCH1*) (*Figure 4A and B*, *Figure 4—figure supplement 1*). In disease, we see the emergence of several stromal region interactions, particularly reticular regions (*Figure 4A*), whereas in health these interactions are downregulated (*Figure 4A*). Furthermore, we see a downregulation of endothelial–subepithelial interactions in health (*VWF-LRP1, MMP9-LRP1*), and in disease, the junctional epithelium shows pro-inflammatory interactions with the adjacent stroma (*NRG1-EGFR, LRIG1-EGFR, TGFA-EGFR*). Collectively, these findings suggest that stromal/immune/endothelial interactions are dominant in disease, whereas structural non-immune interactions shape tissue physiology. Understanding interactions may enable therapeutic modulation to promote immune tolerance in chronic inflammation.

## Spatial location of oral mucosa single cells in health and chronic inflammation

Upon further examination of the oral mucosa spatial features, clusters revealed high cell type complexity. Thus, to dissect cell type composition, we first integrated our previously generated scRNA-seq dataset with the spatial transcriptome data (*Figure 5—figure supplement 1*). This localised

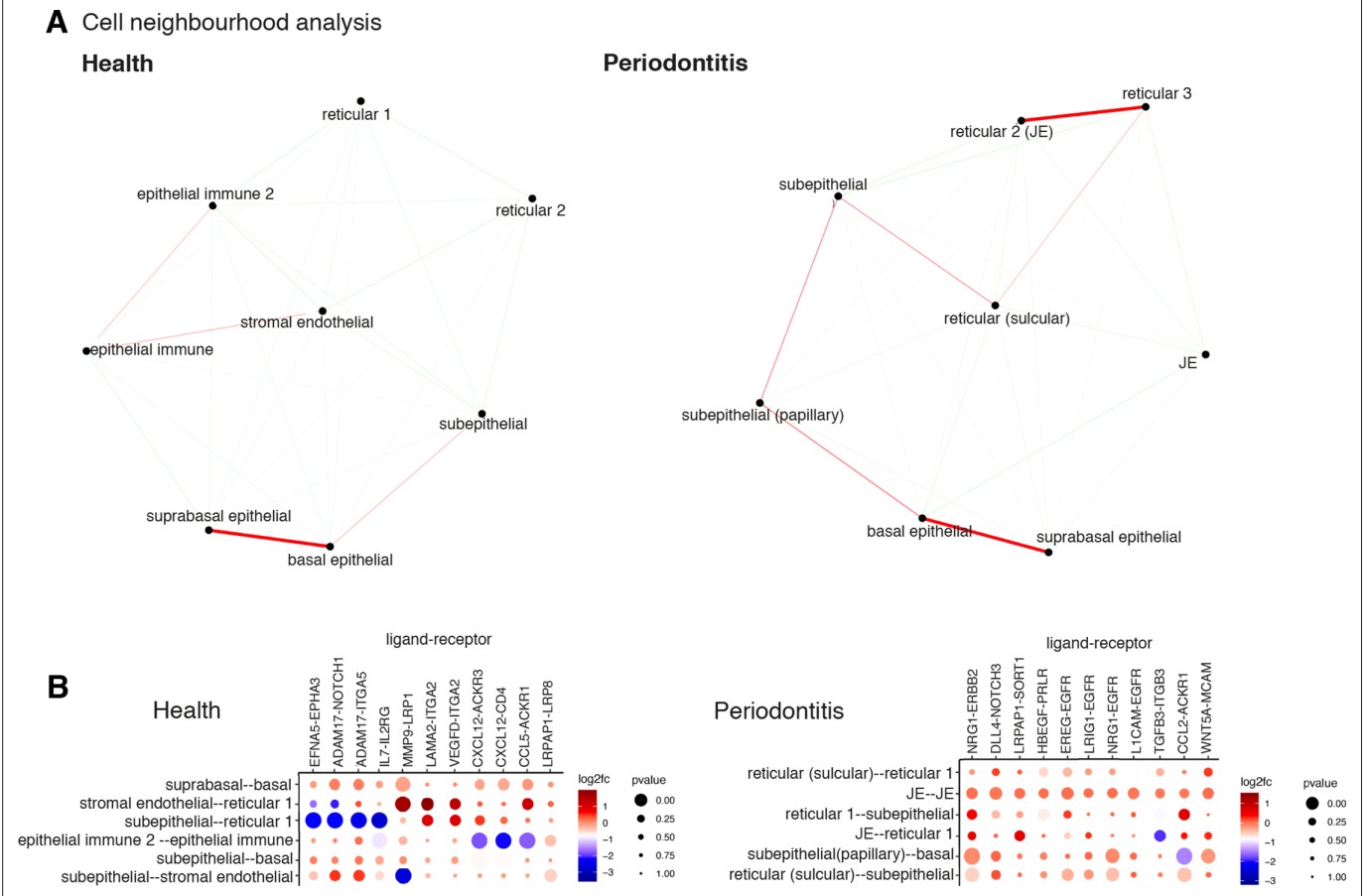

**Figure 4.** Cell neighbourhood analyses in health and disease. (**A**) Network representation of the pairwise interacting regions identified by Giotto Analyzer (**Dries et al., 2021**), whereby it evaluates the enrichment of the frequency that each pair of regions is proximal to each other. Enriched interactions are depicted in red. We observed strongest interactions within epithelial layers in health, whereas in disease stromal regions interactions emerge. (**B**) Dotplot for ligand–receptor pairs that exhibit differential cell–cell communication scores due to spatial cell–cell interactions. The size of the dot is correlated with the adjusted p value, and the colour indicates increased (red) or decreased (blue) activity.

The online version of this article includes the following figure supplement(s) for figure 4:

**Figure supplement 1.** Cell neighbourhood communication analyses.

well-characterised cell types, such as proliferative epithelial cells mapped at the basal epithelial layer, and suprabasal keratinocytes in the suprabasal layer. Next, we used our previously generated high-resolution clustering to map well-characterised cell types to understand whether this strategy could reliably map fine-grained or less characterised cell types. In brief, we examined the expression of cell type-specific marker genes at the spot level and after enhancement. We aggregated the expression of marker genes within each cell type from the literature (*Caetano et al., 2021*; *Huang et al., 2021*; *Williams et al., 2021*) by summing their log-normalised expression (*Figure 5A and B*, *Figure 5—figure supplement 2*). We observed that this method accurately maps oral mucosa cell types, and we were able to detect cell types that went undetected in our reference scRNA-seq dataset, such as melanocytes (*Figure 5A*). Importantly, we observed a substantial increase in endothelial cell expression and distribution in the oral subepithelial and reticular stroma, which contradicts our scRNA-seq analysis alone emphasising the need to complement single-cell analyses with other data modalities. We then applied this method to understand immune cell composition, organisation, and abundance patterns. Again, we were able to detect fine-grained immune cell types that went undetected in our reference dataset, such as γδ T cells and T$_{regs}$ (*Figure 5B*, *Figure 5—figure supplement 2*). Overall, we observed an increase in most immune cell types, particularly of macrophages and γδ T cells. The latter were found in sulcular and junctional epithelial regions. B cells differences were less evident; however, this was expected as these disease samples had a mild inflammatory phenotype to retain

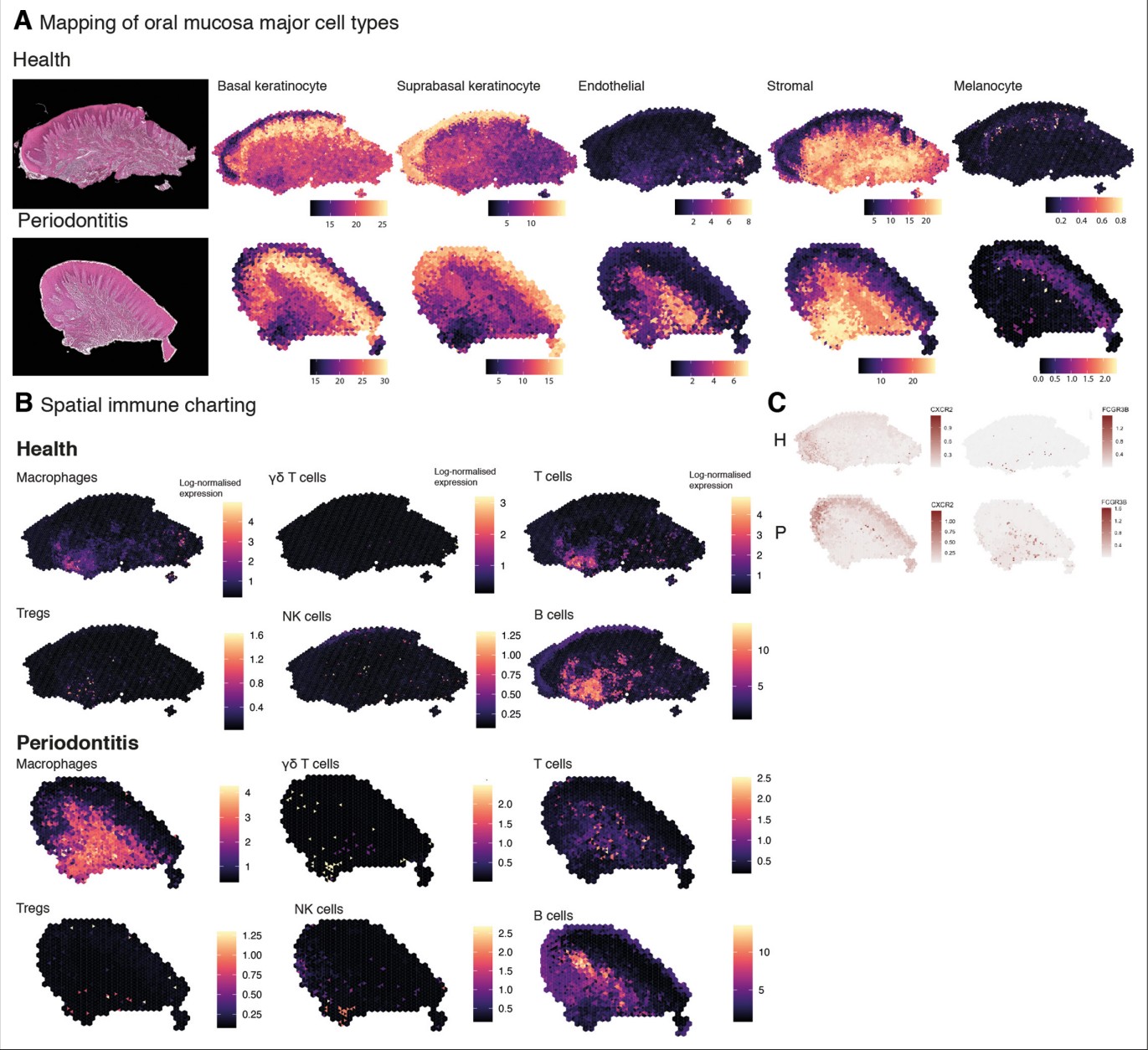

**Figure 5.** Mapping of oral mucosa major cell types across conditions. (**A**) Spatial expression of genes encoding for main human oral mucosa cell types. We selected the top differentially expressed genes for each cell type and predicted their expression at subspot resolution xgboost in BayesSpace. H&E staining shown here for clarity. (**B**) Spatial mapping of immune cell types showing overall increase in disease. (**C**) Neutrophils mapping using expression of *CXCR2* and *FCGR3B*.

The online version of this article includes the following figure supplement(s) for figure 5:

**Figure supplement 1.** Validation of ST method by comparison of human oral mucosa tissue spots with histological landmarks and integration analyses with single-cell RNA sequencing cell type annotations.

**Figure supplement 2.** Cell type mapping for all biological/technical replicates used in ST analyses.

tissue architecture integrity for analysis. Flow cytometry studies have reported that there is a significant recruitment of neutrophils in disease (*Dutzan et al., 2016*) however, in our previous work and in subsequent studies (*Huang et al., 2021*; *Williams et al., 2021*), scRNA-seq was insufficient in characterising these cells. Here, we were able to map these cells and observed a significant increase in disease as reported previously (*Dutzan et al., 2016*).

In short, changes in the immune profile agreed with an initial immune response of a significant infiltration of myeloid cells and lymphocytes, and importantly, composition changes also follow changes in spatial distribution.

## Spatial identification of regional fibroblast subtypes

Next, we used our datasets for mapping fine-grained cell types in the oral mucosa. We focused on the heterogeneity of fibroblasts which remain largely functionally undefined despite their integral role in shaping tissue homeostasis and disease processes (*Davidson et al., 2021*). Single-cell technologies have changed our ability to study the molecular properties of these cells; however, their positional specification remains limited due to difficulties in specifying them since they share many common markers.

To identify regionally enriched fibroblasts, we first annotated and spatially mapped five molecularly distinct fibroblast subtypes that were identified based on our scRNA-seq reference. In brief, we considered fibroblast 1–5 in our initial analysis. We then applied the 'anchor'-based integration from Seurat to merge these cells across all seven Visium datasets to obtain prediction scores for each spot for each class of fibroblast. Only three of the five subtypes mapped into our spatial data; therefore, we used BayesSpace to map all subtypes using Seurat differential gene expression markers data (*Figure 6A*, *Figure 6—figure supplement 1*). The extent of molecular heterogeneity across these subtypes (measured by the number of marker genes) is as expected lower than that across different broad cell types, demonstrating the transcriptionally fine-grained nature of these cell populations. We identified subtypes that were spatially enriched in the reticular (Fb1, Fb3), subepithelial (Fb2), and immune enriched regions (Fb4, Fb5). This mapping largely confirmed previous spatial predictions based on their individual transcriptome (*Figure 6A*). Fibroblast 2 ($GREM1^+$, $SFRP1^+$) showed an increased spatial enrichment in the subepithelial region compared to the other subtypes, and a substantial enrichment in the sulcular stroma, suggestive of a role in epithelial maintenance. To further understand this, we investigated how this population and epithelial cells interact with each other by using a ligand–receptor pipeline (*Browaeys et al., 2020*); this analysis predicted *RSPO1*, *CLEC11A*, and *FLRT2* as top ranked ligands expressed by Fb2 based on target genes expressed by epithelial cells (*Figure 6D*). Thus, in human oral mucosa, Fb2 fibroblasts may be required for epithelial niche maintenance.

Both fibroblasts 4 and 5 localised in immune enriched areas, which confirmed their potential role in immune regulation with enrichment for various cytokine-mediated signalling pathways, IFNγ signalling, and pathways involved in T-cell activation (*Figure 6A and E*, *Figure 6—figure supplement 1*).

Overall, these results follow previous evidence of location-specific transcriptional programmes that reflect the individual functional requirements of the surrounding tissue, and which were shown to be imprinted during development by epigenetic regulation of HOX genes (*Frank-Bertoncelj et al., 2017*).

## Rare pathogenic fibroblasts are location-specific and control lymphocyte recruitment and blood progenitor development

Although there are common mechanisms that fibroblasts use to regulate tissue immunity across diseases, others are unique to a single disease or anatomical location, and consequently defined by their local tissue environment (*Davidson et al., 2021*). Thus, we focused on fibroblast 5, which represented a rare cell population (<1%) in our reference dataset, and which was found markedly increased in disease and barely detected in healthy patients (*Figure 6B*). These fibroblasts were mainly located in regions that co-localised with other immune cell types, and are characterised by expression of *RAC2*, *LCP1,* and *TNFRS11B*, a negative regulator of bone resorption and key regulator of osteoclast activity as previously described (*Figure 6B–E*). Overall, this population shows enrichment of genes involved in 'cytokine production', 'immune response', and 'T-cell activation' (*Figure 6F*). This immune-effector profile is also shown by a higher expression of chemokines such as *CXCL8* and *CXCL10,* and *ALOX5AP*, which encodes arachidonate 5-lipoxygenase-activating protein. To support this, we mapped these in our ST data, and these co-localised in the same immunogenic region as Fb5, macrophages, and B cells (*Figure 6G*), adjacent to the junctional epithelium region. *CXCL8* is known to mediate and guide neutrophil recruitment (*de Oliveira et al., 2013*), whilst also being a potent angiogenic factor (*Heidemann et al., 2003*). *CXCL10,* also known as interferon-gamma-inducible protein 10, is secreted in

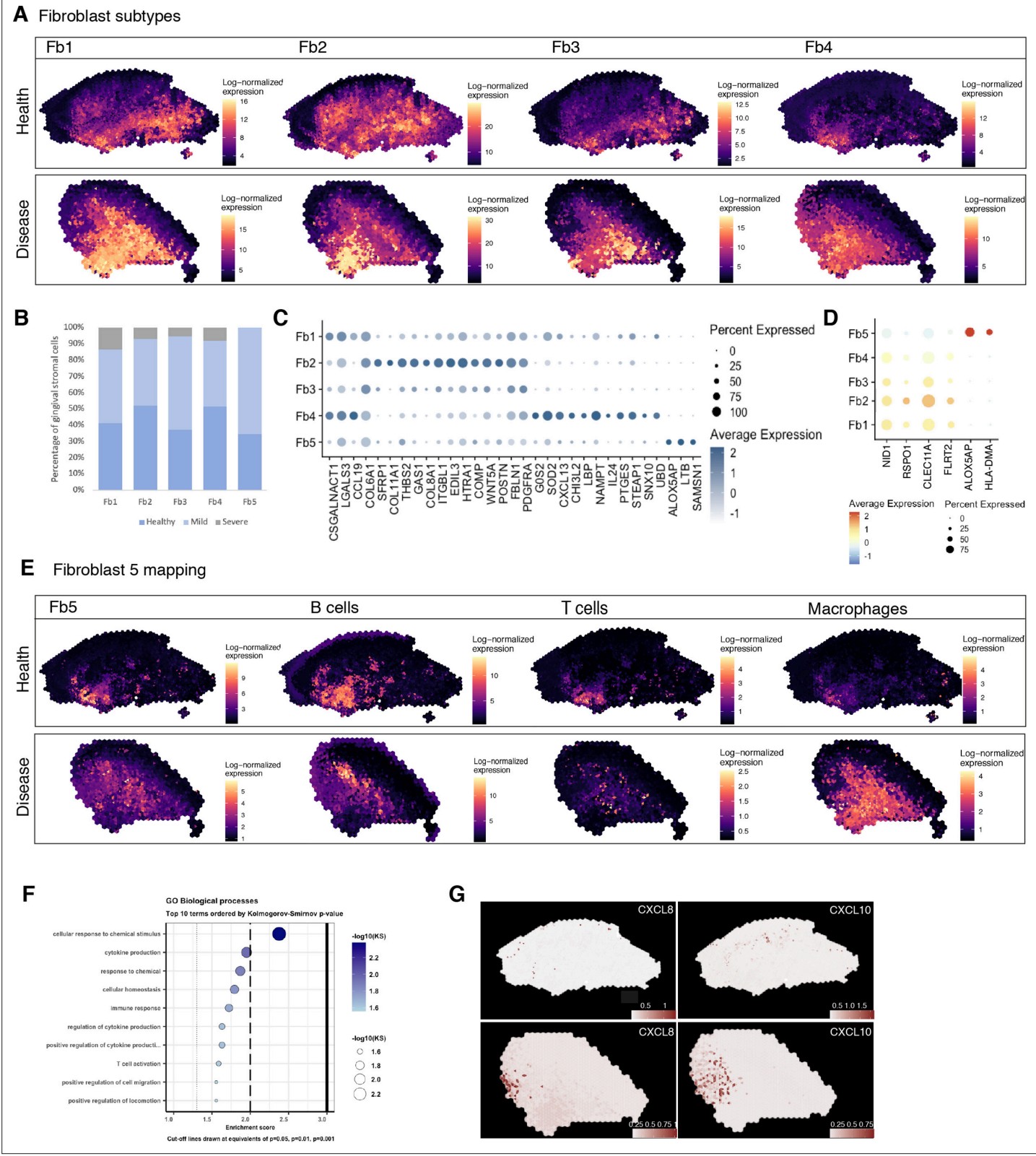

**Figure 6.** Spatial identification of regional fibroblast subtypes. (**A**) Human oral mucosa fibroblast spatial mapping in health and disease. We used the top differentially expressed genes for each cell type and predicted their expression with BayesSpace. (**B**) Percentage of selected fibroblast abundance changes across conditions shown as a bar plot from single-cell RNA sequencing data. (**C**) Dotplot showing markers of fibroblast differentially expressed genes showing percent of expressing cells (circle size) and average expression (colour) of gene markers (rows) across cells (columns). (**D**) Dotplot

*Figure 6 continued on next page*

*Figure 6 continued*

showing top predicted ligands expressed by fibroblast types predicted to modulate the epithelial basal layer. (**E**) Fibroblast 5 spatial mapping and co-localisation with immune cell types, B cells, T cells, and macrophages. (**F**) Gene Ontology biological process term enrichment plot for fibroblast 5. Cluster markers were obtained using the non-parametric two-sided Wilcoxon rank-sum test in Seurat. Gene enrichment analysis was performed with the topGO package in R using the Kolmogorov–Smirnov statistical test. (**G**) Representative spatial mapping of *CXCL8* and *CXCL10* in health and disease showing co-localisation in fibroblast 5 region and increase in disease.

The online version of this article includes the following figure supplement(s) for figure 6:

**Figure supplement 1.** Fibroblast mapping for all biological/technical replicates used in ST analyses.

response to IFN-γ and binds to its receptor *CXCR3* regulating immune responses through recruitment of monocytes/macrophages, T cells, NK cells, and dendritic cells (*Lee et al., 2009*; *Rotondi et al., 2007*). *ALOX5AP* occupies a central role in the production of leukotrienes from arachidonic acid, and recently it was found to regulate blood progenitor formation (*Ibarra-Soria et al., 2018*).

We validated the identity, distribution, and co-localisation of fibroblast 5 with immune cells using multiplex fluorescence ISH (*Figure 7*). We used *RAC2* and *LCP1* to label this population, and *CD14* and *CD79A* for macrophages and B cells, respectively. *In situ* expression patterns validated the spatial transcriptional mapping and are suggestive of a positive-feedback loop, engaging in paracrine interactions with these adjacent immune cells. We hypothesise that similar to other chronic inflammatory conditions, these fibroblasts might help the transition between innate and adaptive immune responses by recruiting and maintaining lymphocyte populations.

Collectively, these results spatially define fibroblast subsets and identify a rare pathogenic fibroblast subset engaged in immune cell recruitment and formation of blood progenitor cells.

## Discussion

Tissue maintenance and repair depend on the integrated activity of multiple cell types (*Kotas and Medzhitov, 2015*). Profiling the human oral mucosa in space is essential to define cell states and signalling pathways in health and oral chronic inflammation. This cellular and tissue reference will improve our understanding of the molecular and cellular deviations occurring in one of the highest prevalent chronic conditions. Furthermore, due to a remarkable wound-healing capacity, understanding this tissue at such resolution might provide new approaches for tissue engineering therapies.

Previously, we provided the first description of the human oral mucosa (*Caetano et al., 2021*) at a single-cell level, demonstrating that cellular remodelling in disease is highly heterogenous, occurring in a cell-specific manner, where populations involved in tissue maintenance are compromised and pro-inflammatory ones persist or expand. Subsequent similar studies were able to support these findings in a larger cohort of patients (*Huang et al., 2021*; *Williams et al., 2021*). Here, we present an integration of dissociated single-cell, spatial transcriptomics, and high-resolution quantitative multiplex imaging to achieve a high-throughput approach to map tissue cell types and cell signalling *in situ*. We asked whether diseased tissue structure drive changes in oral mucosa cell decisions, and whether there is a connection between immune composition and tissue architecture.

We found that the lamina propria can be divided in highly specialised spatial compartments distinctively involved in epithelial maintenance, ECM remodelling, and immune regulation. We discovered pro-inflammatory stromal features involved in neutrophil recruitment in the junctional epithelium region, and an angiogenic stromal signature in the sulcus. We were also able to provide the first molecular mapping of the three distinct epithelia that compose the masticatory mucosa, and how the underlying lamina propria seems to be specified to each epithelial compartment. Of these, the junctional epithelium attaches directly to the tooth surface and its destruction leads to inflammation and alveolar bone loss; thus, understanding how this compartment is maintained remains a goal in the field. Of interest, we found *SOX9* expression restricted to the junctional epithelium. Recent lineage-tracing studies have suggested the presence of Wnt-responsive stem cells contributing to tissue renewal (*Tanaka et al., 2021*; *Yuan et al., 2021*), but their identity remains unknown. *SOX9* is known to be expressed in stem cells/progenitors in the gut and skin (*Jo et al., 2014*); thus, we hypothesise that it may label similar populations in the human oral mucosa. Further lineage-tracing studies will be required to test this. Within the junctional epithelium, we were also able to identify a rare proliferative population that we previously identified in our reference dataset, expressing *TOP2A*

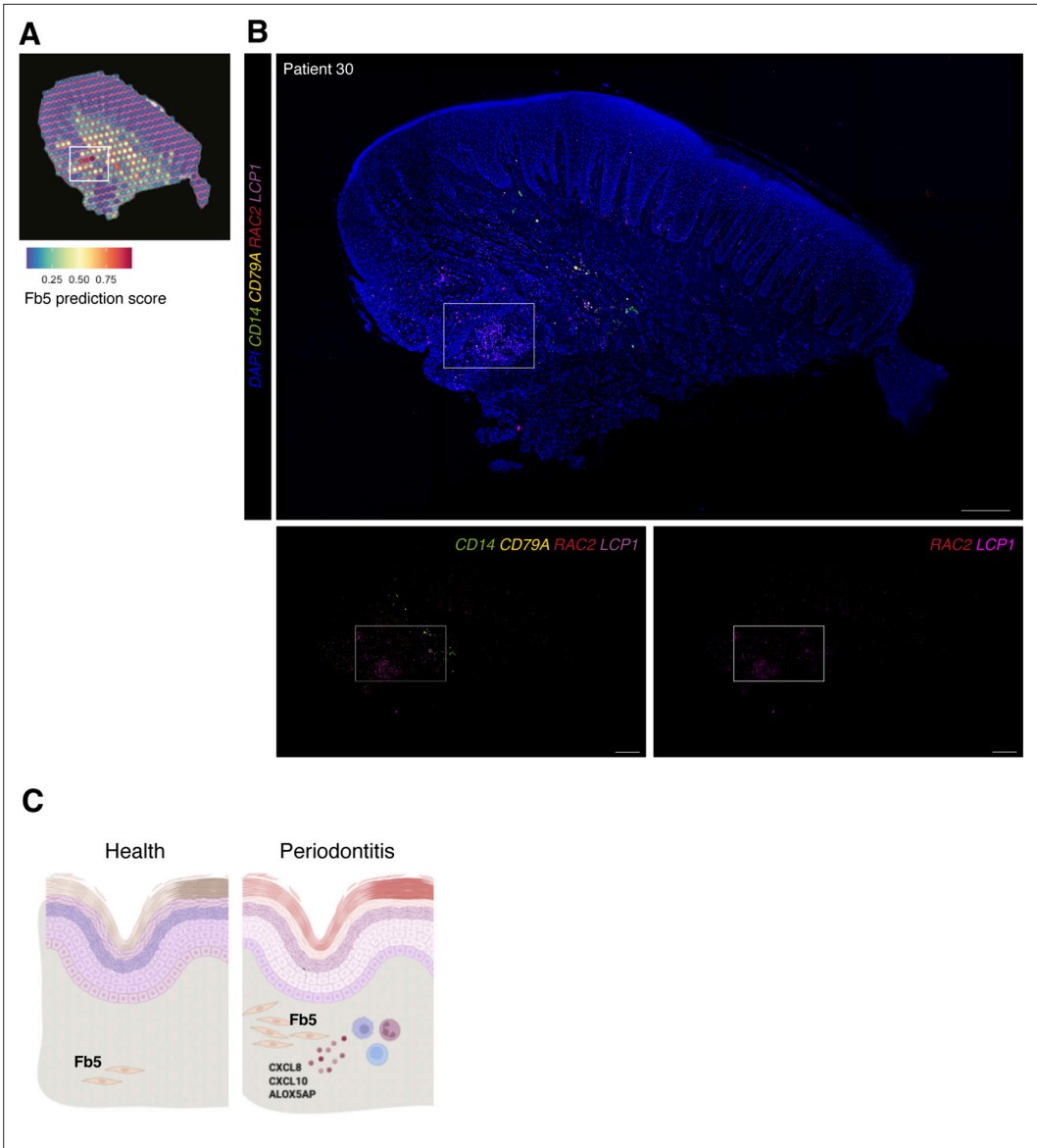

**Figure 7.** *In vivo* validation of fibroblast 5 co-localisation with immune cells. (**A**) Single-cell RNA sequencing integration with spatial data using 'anchor'-based integration workflow in Seurat to enable a prediction score for each spot for fibroblast 5. Based on this prediction score, fibroblast 5 is confirmed spatially restricted. (**B**) Multiplex mRNA *in situ* hybridisation of a representative section of human oral mucosa using specific probes against *RAC2* and *LCP1* (fibroblast 5 markers), *CD14* (macrophages), and *CD79A* (B cells) to validate *in vivo* localisation of fibroblast 5 and immune cell populations. Scale bars: 250 µm. (**C**) Schematic illustration of our proposed model summarising pathogenic fibroblast population role in human oral chronic disease. Created with Biorender.

and *CENPF*. *TOP2A* targets are largely transcribed in embryonic stem cells, and a study demonstrated a critical role in defining stem cell pluripotency and differentiation potential (*Thakurela et al., 2013*). Finally, we attempted to link disease susceptibility with phenotype by mapping known risk genes, and surprisingly, identified a restricted expression in the junctional epithelium for *CDKN2B*, *IL1A*, *IL1B*, and *VAMP8*, suggesting that junctional epithelium dysfunction underlies periodontitis pathogenesis.

We provided a candidate *in situ* cell–cell communication network showing the emergence of a stromal–immune–endothelial axis in disease, which also revealed an increase in WNT receptor

expression in basal–subepithelial interactions. In the healthy mucosa, we detected increased interactions between basal and suprabasal epithelial cells governed by *ADAM17* and *NOTCH1*. The oral epithelium exhibits a remarkable epithelial turnover; however, our understanding of oral epithelial interactions and how epithelial differentiation is regulated is limited.

While recent studies have studied oral epithelial stem cell/progenitors' heterogeneity and their role in tissue homeostasis (*Byrd et al., 2019*; *Jones et al., 2019*), much less is known about mesenchymal cells' heterogeneity and role in tissue maintenance. This is explained by a lack of robust markers and the difficulty of successful lineage tracing studies given that these cells are not maintained together. We previously identified five distinct fibroblast populations in the human oral mucosa (*Caetano et al., 2021*) however, we found a small number of transcriptional differences which conditioned a successful mapping of these cells with traditional *in situ* techniques. Furthermore, scRNA-seq data does not capture cell fates but rather cell states with recent evidence suggesting that many fibroblast lineage markers reflect a fibroblast state (*Thompson et al., 2021*). In this study, we were able to map these fine-grained differences and observed distinct spatial enrichment. In addition to their molecular and spatial identities, we can only speculate about their regionalised functions; however, subepithelial fibroblasts similar to our Fb2 population have been reported in other tissues and mechanistically found to support epithelial stem cell maintenance (*Bahar Halpern et al., 2020*; *Kinchen et al., 2018*; *Shoshkes-Carmel et al., 2018*).

Finally, we focused on mapping a rare pro-inflammatory fibroblast subtype that we previously discovered expanded in disease. Here, we found that it co-localises with several immune cell populations and specifically expresses *CXCL8* and *CXCL10*. *CXCL8* is a major activating chemotactic factor of human neutrophils (*Luster, 1998*; *Zlotnik and Yoshie, 2012*) and a potent angiogenic factor (*Heidemann et al., 2003*); *CXCL10* was initially identified as a chemokine secreted by several cell types in response to IFN-γ, regulating immune responses through the recruitment of leukocytes, including T cells and macrophages, and activation of toll-like receptor 4 (TLR4). More recently, it was found as a critical regulator of intestinal immunity in an enteric glia axis (*Progatzky et al., 2021*). This pathogenic population also highly expresses *ALOX5AP*, which is known to control leukotriene production from arachidonic acid. Interestingly, it was recently found to also modulate formation of early blood progenitor cells (*Ibarra-Soria et al., 2018*). Periodontal lesions are associated with activation of pathological angiogenesis and a high number of newly formed blood vessels that facilitate local immune cell migration. It will be intriguing to decipher how the leukotriene pathway promotes the formation of blood progenitor cells in the oral mucosa; this pathway will be easier to exploit in a translational setting because of the ready availability of small-molecule agonists and antagonists. Collectively, our data suggest the emergence of a rare and spatially restricted pathogenic fibroblast population responsible for neutrophil and lymphocyte recruitment at pocket sites, and with angiogenic properties. We suggest that this population is responsible for irregular chemokine gradients, resulting in the capture and accumulation of inflammatory cells. Thus, we speculate that this fibroblast may help in the transition from innate to adaptive immunity and in the development of persistent inflammation characteristic of periodontitis, as suggested in other chronic inflammatory disease (*Davidson et al., 2021*). Once fibroblasts are exposed to persistent inflammation, they sustain lymphocyte retention which may reflect their inability to return to a homeostatic state. In fact, these cells are resistant to apoptosis and retain pathogenic features in the absence of disease (*Whitaker et al., 2013*). Discreet pathogenic populations are also present in other chronic inflammatory diseases (*Croft et al., 2019*; *Kinchen et al., 2018*; *Smillie et al., 2019*), which are largely defined by their anatomical location. The relationship between tissue specificity and pathological cues in shaping fibroblast phenotype and heterogeneity in disease is still a source of much debate; however, given that fibroblasts are embedded in a network of immune, epithelial, and endothelial populations, they adopt properties and address codes to complex paracrine cues from surrounding cells. Thus, our integrative map of cellular profiles will serve as an essential reference for the study of mucosal homeostasis and chronic inflammatory disease, as well as a blueprint for tissue engineering approaches.

# Materials and methods

## Key resources table

| Reagent type (species) or resource | Designation | Source or reference | Identifiers | Additional information |
|---|---|---|---|---|
| Biological sample (human) | Oral mucosa biopsies | Periodontology department, King's College London | | |
| Antibody | Anti-KRT14 (mouse monoclonal) | Abcam | Cat# ab7800; RRID:AB_306091 | IHC (1:100) |
| Antibody | Anti-KRT76 (rabbit polyclonal) | Atlas Antibodies | Cat# HPA019656 | IHC (1:100) |
| Commercial assay or kit | Visium Gene Expression Slide Kit, 4 rxns | 10X Genomics | 10X-100187 | |
| Commercial assay or kit | Visium Gene Expression Slide Kit, 4 rxns | 10X Genomics | 10X-100338 | |
| Commercial assay or kit | RNAscope Multiplex Fluorescent V2 | ACD | Cat# 323100 | |
| Commercial assay or kit | RNAscope 4-Plex Ancillary Kit for Multiplex Fluorescent Kit V2 | ACD | Cat# 323120 | |
| Commercial assay or kit | RNAscope Target Probes (Made-to-Order C4 Probes). | ACD | | |
| Software, algorithm | Space Ranger (version 1.0.0) | 10X Genomics | https://support.10xgenomics.com/spatial-gene-expression/software/pipelines/latest/installation | |
| Software, algorithm | Seurat version 4.0 | R Bioconductor | RRID:SCR_007322; https://satijalab.org/seurat/ | |
| Software, algorithm | Enrichr | *Chen et al., 2013* | RRID:SCR_001575 | |
| Software, algorithm | NicheNet | GitHub, *Browaeys et al., 2020* | https://github.com/saeyslab/nichenetr | |
| Software, algorithm | BayesSpace | R Bioconductor | https://www.bioconductor.org/packages/release/bioc/html/BayesSpace.html | |
| Software, algorithm | Giotto | GitHub, *Dries et al., 2021* | https://rubd.github.io/Giotto_site/ | |
| Software, algorithm | topGO | R Bioconductor | https://bioconductor.org/packages/release/bioc/html/topGO.html | |

## Tissue handling

Human oral mucosa samples were obtained from consenting patients undergoing routine periodontal surgical procedures (Department of Periodontology, Guy's Hospital, King's College London). All samples were collected and processed in compliance with the UK Human Tissue Act (Human Tissue Authority #203019), ethically approved by the UK National Research Ethics Service (Research Ethics Committee 17/LO/1188). Written informed consent was received from participants prior to inclusion in the study. Cohort inclusion criteria for all subjects were absent history of relevant medical conditions, no use of medication, no use of nicotine or nicotine-replacement medications, no pregnancy, and breast feeding.

## Sample information

An overview of all samples used in the study is deposited in the supplementary materials and details such as age, gender, location, and disease state where appropriate. No randomisation or blinding of samples was performed. A power-calculation was not performed prior to study as samples were processed based on tissue quality, anatomical integrity, location, and availability.

Healthy controls included crown lengthening procedures, and periodontitis patients, pocket reduction surgeries. Patients with periodontitis had tooth sites with probing depth ≥6 mm and bleeding on probing. Patients used as controls showed no signs of periodontal disease, with no gingival/periodontal inflammation, a probing depth ≤3 mm, and no bleeding on probing.

Patient 7. Gender: female. Age band: 41–65. Chronic periodontitis with previous history of non-surgical treatment (mild). Site: buccal gingival margin.

Patient 30. Gender: female. Age band: 41–65. Chronic periodontitis with previous history of non-surgical treatment (mild). Site: buccal gingival margin.

Patient 43. Gender: male. Age band: 41–65. No history of periodontal disease. Site: buccal gingival margin.

Patient 35. Gender: male. Age band: 41–65. Chronic periodontitis with previous history of non-surgical treatment (mild). Site: buccal gingival margin.

Patient 36. Gender: female. Age band: 41–65. No history of periodontal disease. Site: buccal gingival margin.

Patient 41. Gender: female. Age band: 41–65. Chronic periodontitis with previous history of non-surgical treatment (mild). Site: buccal gingival margin.

Patient 45. Gender: female. Age band: 41–65. No history of periodontal disease. Site: palatal gingival margin.

## Spatial transcriptomics

Human oral mucosa tissue was collected immediately after clinical surgery and fixed overnight in 4% neutral buffered formalin for the FFPE protocol. Tissues underwent three 5 min washes in PBS at room temperature followed by dehydration washes in increasing ethanol concentrations. After dehydration, tissue was processed using a Leica ASP300 Tissue Processing for 1 hr. Tissues were then embedded in paraffin and stored at 4°C. For the frozen method, samples were collected immediately after clinical surgery (within 30 min), snap-frozen in O.C.T. compound using liquid nitrogen, and subsequently stored at –80°C in an air-tight container. Frozen blocks were tested for RNA quality with RIN > 8 for frozen tissues (RNA pico, Agilent) and three tissue optimisation experiments (10X Genomics, Visium Spatial Tissue Optimisation, Rev A) were performed with imaging of fluorescence footprint on a Nikon Eclipse Ti2 microscope and image analysis performed in Fiji (ImageJ v2.0.0). We identified 12 min as optimum permeabilisation time. FFPE blocks were tested for RNA quality with DV200 > 50% and a tissue adhesion test was performed.

Samples were then processed for full ST experiment as per the manufacturer's instructions (10X Genomics, Visium Spatial). Frozen samples were cut in a pre-cooled cryostat at 15 μm thickness onto four 6.5 mm × 6.5 mm capture areas with 5000 oligo-barcoded spots. For some samples (patients 43 and 45), n = 2 technical replicates were performed. Slides then underwent fixation and H&E staining with immediate imaging on Nanozoomer S60 (Hamamatsu) at ×40 magnification. Tissue underwent permeabilisation with proprietary enzyme (12 min), reverse transcription, and second-strand synthesis performed on the slide with cDNA quantification undertaken with qRT-PCR using KAPA SYBR FAST-qPCR kit (KAPA Systems) and analysed on the QuantStudio 7-Flex system (Thermo Fisher). FFPE samples were cut at the standardised 5 μm thickness onto four 6.5 mm × 6.5 mm capture areas. Slides then underwent deparaffinisation and H&E staining according to the manufacturer's instructions (10X Genomics, Visium Spatial) with immediate imaging on Nanozoomer S60 (Hamamatsu) at ×40 magnification. Decrosslinking was performed to release RNA that was sequestered by the formalin fixing, followed by probe hybridisation, ligation, release, and extension according to the manufacturer's instructions.

Following library construction as per the manufacturer's instructions ST libraries were quantified using the KAPA-Illumina PCR quantification kit (KAPA Biosystems) and pooled at 4 nM concentration with a sample ratio corresponding to the surface area of tissue coverage obtained from the H&E imaging. Pooled libraries were sequenced on a NextSeq (Illumina) using 150 base-pair paired-end dual-indexed set up loaded. Two slides were sequenced to a manufacturer-recommended depth of approximately 50,000 reads per tissue covered spot.

## Immunohistochemistry and *in situ* hybridisation

For IHC, paraffin-embedded blocks were cut at a thickness of 12 μm onto slides. In short, slides were dewaxed in Neo-Clear twice for 10 min and rehydrated in a series of decreasing ethanol volumes as

described above. Heat-induced epitope retrieval was performed with sodium citrate buffer (pH 6) in a Decloaking chamber NXGEN (Menarini Diagnostics) for 3 min at 110°C. Slides were cooled to room temperature before blocking for 1 hr at room temperature in Blocking Buffer (0.2% BSA, 0.15% glycine, 0.1% TritonX in PBS) with 10% goat or donkey serum depending on the secondary antibody used. Primary antibodies (anti-KRT14 ab7800; anti-KRT76 HPA019656) were diluted in blocking buffer with 1% of the respective blocking buffer and incubated overnight at 4°C. The following day, slides were washed three times in PBST and incubated with the respective secondary antibodies diluted 1:500 in 1% blocking buffer for 1 hr at room temperature. Slides were mounted with Citifluor AF1 mountant media (Citifluor Ltd., AF1-100) and cover slipped for microscopy. Slides were put to dry in a dry chamber that omitted all light, and kept at 4°C.

Multiplex ISH was performed using kits and probes from Advanced Cell Diagnostics (ACD) which used oligonucleotide probes to RNA targets (RNAscope 4-plex Ancillary Kit for Multiplex Fluorescent Reagent Kit v2; Cat# 323120) as per the manufacturer's instructions with incubations steps of processed paraffin-embedded tissue performed in HybEz oven (ACD, Cat# 321461). Akoya Biosciences Opal fluorophores were used for detection of fluorescent signals.

Fluorescent staining was imaged with a TCS SP5 confocal microscope (Leica Microsystems) and Leica Application Suite Advanced Fluorescence (LAS-AF) software. Images were collected and labelled using Adobe Photoshop 21.1.2 software and processed using Fiji (*Schindelin et al., 2012*).

## Raw sequencing data processing

Raw sequencing data was converted to from bcl to fastq format using Illumina bcl2fastq software. Raw sequencing reads were quality checked using FastQC software.

Human hg38 reference genome analysis set was downloaded from the University of California Santa Cruz (UCSC) ftp site (*Kuhn et al., 2013*). Human hg38 reference genome Emsembl gene annotations were obtained using the UCSC Table Browser Tool (*Karolchik et al., 2004*).

Spaceranger (version 1.0.0) software from 10X Genomics was used to process, align, and summarise UMI counts against hg38 human reference genome for each spot on the Visium spatial transcriptomics array.

## Spatial transcriptomics data analysis

Raw UMI counts spot matrices, imaging data, spot-image coordinates, and scale factors were imported into R using the Seurat package (*Butler et al., 2018*; *Stuart et al., 2019*). Raw UMI counts were normalised using regularised negative binomial regression (SC Transform; *Hafemeister and Satija, 2019*) to account for variability in total spot RNA content. Dimensionality reduction was performed using PCA, and for each slide, scree plots were examined to determine the optimum number of principal components to use in downstream clustering analyses.

Clustering was performed using Louvain clustering algorithm as before (resolution = 0.5) and clusters were visualised using UMAP algorithm as before. Cluster distributions were visualised in spatial context over H&E images with spot size scaling factor of 2–3.5 used throughout. Our previous scRNA-seq dataset (GSE152042) was integrated via FindTransferAnchors and TransferData functions in Seurat to predict spot content for each slide. To further increase data resolution at a subspot level, we applied the BayesSpace package (*Zhao et al., 2021*). Briefly, Visium datasets processed with Space Ranger were loaded directly via the readVisium function. In all datasets, raw gene expression counts were log transformed and normalised using the spatialPreprocess function and PCA was then performed on the top 2000 most highly variable genes. We used qTune and qPlot to determine the optimum number of principal components. Clustering was performed using a Bayesian model with a Markov random field. To enhance the resolution of this clustering map, we segmented each spot into subspots with the spatialEnhance function and again leveraged spatial information using the Potts model spatial prior; it segments each Visium spot into six subspots and computes enhanced resolution PC vectors using a regression algorithm.

## Marker gene detection and differential expression analyses

For all marker gene expression, we used R packages Seurat and BayesSpace. Briefly, for each identified cluster, we compared the cells within that cluster versus all cells. For visualising and thresholding cell type specificity, we calculated gene AUC scores for all cell types using area under the curve

analysis for each gene as implemented in Seurat FindMarkers function (test = ROC). Gene expression enhancement was implemented using the enhanceFeatures function in BayesSpace. We selected sets of marker genes of each cell type and predicted their expression at subspot resolution using xgboost.

## Spatially informed ligand–receptor analysis

To perform neighbourhood analyses we used the R package Giotto (*Dries et al., 2021*). It incorporates ligand–receptor information from existing databases (*Ramilowski et al., 2015*) and calculates the increased spatial co-expression of gene pairs in neighbouring cells from two cell types. Then, it estimates which ligand–receptor pairs might be used often for communication between interacting cells. This is applied with the function spatCellCellcom.

## NicheNet analysis

This analysis predicts which ligands produced by a sender cell regulate the expression of receptors/target genes in another (receiver) cell. We followed the open-source R implementation available at GitHub (https://github.com/saeyslab/nichenetr). For differential expression, we used FindMarkers function in Seurat to generate average logFC values per cell type. For *Figure 6D*, we assigned stromal cells from the scRNA-seq reference dataset as 'sender cells' and epithelial populations as 'received cells'.

## Gene Ontology enrichment analysis

Gene Ontology enrichment analyses were performed using using Enrichr (*Chen et al., 2013*) on the top 200 differentially expressed genes (adjusted p-value<0.05 by Wilcoxon rank-sum test). GO terms shown are enriched at false discovery rate (FDR) < 0.05. topGO R package available from Bioconductor was also applied with the Kolmogorov–Smirnov test (*Alexa and Rahnenfuhrer, 2022*).

## Code availability

Code to reproduce the analysis is available at https://github.com/anacaetano/human-oral-mucosa-spatial/.

## Study approval

Informed consent in writing before their participation in this study was obtained from each subject in compliance with the UK Human Tissue Act (Human Tissue Authority #203019) and ethically approved by the UK National Research Ethics Service (Research Ethics Committee 17/LO/1188).

# Acknowledgements

We thank all the patients who contributed to this study, the support of our Periodontology MClin-Dent students, GSTT nursing staff, and Dr Pegah Heidarzadeh Pasha at Guy's Hospital. We thank all CCRB laboratory technicians for their support, especially Dr Alasdair Edgar for tissue processing. We acknowledge support of the BRC Genomics cores at Guy's Hospital for their services. We acknowledge Dr Cynthia Andoniadou lab for support with the RNAscope experiments. We thank Dr Inês Sequeira Lab for the antibody Keratin 76. The research described was supported by the BBSRC Industrial CASE Studentship (Grant Ref: BB/P504506/1) and National Institute for Health Research's Biomedical Research Centre based at Guy's and St Thomas' NHS Foundation Trust and King's College London. The views expressed are those of the authors and not necessarily those of the NHS, the National Institute for Health Research, or the Department of Health. This work was funded by Czech Science Foundation, project 21-21409S and Unilever in the form of a research grant awarded to PTS.

# Additional information

### Competing interests

Eleanor M D'Agostino: is an employee of Unilever Plc. The authors state no conflict of interest. The other authors declare that no competing interests exist.

## Funding

| Funder | Grant reference number | Author |
|---|---|---|
| Biotechnology and Biological Sciences Research Council | BB/P504506/1 | Ana J Caetano |
| NIHR Biomedical Research Centre Guy's and St Thomas' NHS Foundation Trust and King's College London | | Paul T Sharpe |
| Czech Science Foundation | 21-21409S | Paul T Sharpe |

The funders had no role in study design, data collection and interpretation, or the decision to submit the work for publication.

## Author contributions

Ana J Caetano, Conceptualization, Data curation, Formal analysis, Validation, Investigation, Visualization, Methodology, Writing – original draft, Writing – review and editing; Yushi Redhead, Formal analysis, Visualization, Methodology, Writing – review and editing; Farah Karim, Validation, Methodology, Writing – review and editing; Pawan Dhami, Shichina Kannambath, Rosamond Nuamah, Resources, Data curation, Software, Writing – review and editing; Ana A Volponi, Resources, Formal analysis, Supervision, Methodology, Writing – review and editing; Luigi Nibali, Resources, Data curation, Writing – review and editing; Veronica Booth, Resources, Validation, Writing – review and editing; Eleanor M D'Agostino, Conceptualization, Funding acquisition, Investigation, Writing – review and editing; Paul T Sharpe, Conceptualization, Formal analysis, Supervision, Funding acquisition, Writing – original draft, Project administration, Writing – review and editing

## Author ORCIDs

Ana J Caetano 
Ana A Volponi 
Paul T Sharpe 

## Decision letter and Author response

Decision letter https://doi.org/10.7554/eLife.81525.sa1
Author response https://doi.org/10.7554/eLife.81525.sa2

# Additional files

## Supplementary files

• MDAR checklist

## Data availability

All data generated or analysed during this study are included in the manuscript and supporting file; Source Data files have been provided.

The following dataset was generated:

| Author(s) | Year | Dataset title | Dataset URL | Database and Identifier |
|---|---|---|---|---|
| Caetano AJ | 2022 | Mapping the Spatial Dynamics of the Human Oral Mucosa in Chronic Inflammatory Disease | https://www.ncbi.nlm.nih.gov/geo/query/acc.cgi?acc=GSE206621 | NCBI Gene Expression Omnibus, GSE206621 |

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
