## [Editor Report]

The work by Caetano et al. describes the changes caused by periodontal inflammation in terms of tissue structure through integrating multi-omics techniques and fluorescence in situ hybridization. They defined highly specialized epithelial and stromal compartments and spatially mapped a rare pathogenic fibroblast population likely responsible for lymphocyte recruitment and angiogenesis. They also compared the genes with altered expression in gingival inflammation with the ones highlighted from the GWAS analysis related to patients with periodontitis which contributes to generating new hypotheses for future studies.

---

## [Decision Letter]

**Decision letter after peer review:**

Thank you for submitting your article "Mapping the Spatial Dynamics of the Human Oral Mucosa in Chronic Inflammatory Disease" for consideration by *eLife*. Your article has been reviewed by 2 peer reviewers, one of whom is a member of our board of Reviewing Editors, and the evaluation has been overseen by Mone Zaidi as the Senior Editor. The reviewers have opted to remain anonymous.

Essential revisions:

We believe it is important to address the comments of all reviewers, specifically the following:

1. To improve the resolution, the authors introduced a method that addresses improving the resolution by combining more information from the neighbour structure and the existing database. This raises the question of whether the lack of previous gingival tissue spatial transcriptome sequencing results weakens the reliability of this method. Does it miss the identification of some gingival tissue-specific cells? Is the failure to match two populations of fibroblasts between single-cell sequencing and spatial transcriptome sequencing of gingival tissue fibroblasts related to this?

2. Although the authors did the identification of the captured tissues, the results seem to require more analysis. Take Figure 5A as an example, there is a clear overlap between endothelial cells and basal cells. In addition, it is suggested that the authors indicate the specific location of the 10 clusters of cells in Figures 1D and 2C.

3. Healthy gingival tissues were taken from seven individuals. It would be helpful to know whether there was significant individual variation in cell clusters among these samples. Furthermore, regarding gingival tissue collected from patients with periodontitis, were these tissue samples collected from similar anatomical regions/locations?

4. In healthy gingiva, the authors identified dominant interactions between suprabasal and basal epithelial cells. The authors should comment on the biological significance of these interactions.

5. Are those rare pro-inflammatory fibroblasts close to junctional epithelium?

6. The organization of the cell populations under healthy conditions and periodontitis is very different based on the data shown in Figure 1D and Figure 2C. The authors should compare these populations more clearly and explain the reason behind this.

7. There are ten cell populations shown in Figure 1F and Figure 2D but the interactions highlighted in Figure 4 only have six groups. Could the authors explain this?

8. The heatmap including the differential gene expression profile needs to be shown for the fibroblasts annotated in Figure 6, in addition to the Dot plot. Moreover, the distribution of Fb1-Fb4 is quite similar under diseased conditions. What is the reason behind this?

*Reviewer #1 (Recommendations for the authors):*

1. The results of "Molecular spatial reconstruction of the human oral mucosa in health" (Figure 1) are suggested to be validated using FISH or IHC. In addition, the authors need to describe the sample size of the study more clearly. Figure 1A shows a sample size of 7, but the main text describes 5 healthy samples, and 4 periodontitis samples are included.

2. Figure 2E shows the markers for each tissue compartment, and which of these 41 markers need to be tagged as pro-inflammatory genes to corroborate the authors' conclusion that "we also observed several disease-related changes in the stromal compartment. For example, the overall upregulation of pro-inflammatory genes."

3. The authors cited a reference (Nibali et al., 2019) to analyze the transcriptional defects that might drive the process of periodontitis, but no specific periodontitis-associated genes were given in this article. In addition, the "21 known disease gene" used in the article needs to be shown in the supplemental table. Since it is not known which 21 known disease genes were examined by the authors, it is not possible to determine "In summary, our analysis suggests that dysfunctional junctional epithelium might be a major driver of periodontal disease pathogenesis."

4. In Figure 2E, the image that validates the spatial transcriptome results, the results of KTR76 do not seem to match

5. The authors concluded that "KRT13 and KRT4 being restricted to the sulcular epithelium and KRT13 specific to the junctional epithelium", but the results showed that KRT13 was not specifically distributed in the epithelium. The same was observed for *SOX9*.

6. There are several errors in the Figures in the section "Effect of tissue microenvironments on epithelial identity and dynamics". For example, Figure 3B does not provide the results of FOXP2 and NEFM expression and Figure 2G has no results related to the signaling pathway.

7. Microenvironments are not well characterized by only looking at gene expression levels in the section "Effect of tissue microenvironments on epithelial identity and dynamics".

8. According to the legend of Figure 4A, the abundant interactions are depicted in red. However, the results labeled in red in the healthy samples are not epithelial and basal epithelial layers which are not consistent with the authors' description that "in health we observed a major interaction between the epithelial and basal epithelial layers."

9. Is there any other explanation for "we observed a substantial increase in endothelial cell expression and distribution in the oral subepithelial and reticular stroma, which contradicts our scRNA-seq analysis alone emphasizing the need to complement single-cell analyses with other data modalities."?

10. The markers of cells in Figures 5A and B are critical, but the authors do not provide this information.

11. The raw spatial sequencing data obtained from patients used in this study is deposited under GSE206621. Are the results of the healthy control tissue also in this dataset?

*Reviewer #2 (Recommendations for the authors):*

There are some specific questions that will need to be addressed before this manuscript is ready for publication.

1. Healthy gingival tissues were taken from seven individuals. It would be helpful to know whether there was significant individual variation in cell clusters among these samples. Furthermore, regarding gingival tissue collected from patients with periodontitis, were these tissue samples collected from similar anatomical regions/locations?

2. In healthy gingiva, the authors identified dominant interactions between suprabasal and basal epithelial cells. The authors should comment on the biological significance of these interactions.

3. Are those rare pro-inflammatory fibroblasts close to junctional epithelium?

4. The organization of the cell populations under healthy conditions and periodontitis is very different based on the data shown in Figure 1D and Figure 2C. The authors should compare these populations more clearly and explain the reason behind this.

5. There are ten cell populations shown in Figure 1F and Figure 2D but the interactions highlighted in Figure 4 only have six groups. Could the authors explain this?

6. The heatmap including the differential gene expression profile needs to be shown for the fibroblasts annotated in Figure 6, in addition to the Dotplot. Moreover, the distribution of Fb1-Fb4 is quite similar under diseased conditions. What is the reason behind this?

---

## [Author Response]

Essential revisions:We believe it is important to address the comments of all reviewers, specifically the following:1. To improve the resolution, the authors introduced a method that addresses improving the resolution by combining more information from the neighbour structure and the existing database. This raises the question of whether the lack of previous gingival tissue spatial transcriptome sequencing results weakens the reliability of this method. Does it miss the identification of some gingival tissue-specific cells? Is the failure to match two populations of fibroblasts between single-cell sequencing and spatial transcriptome sequencing of gingival tissue fibroblasts related to this?

Thank you for raising these concerns. We don’t think that the lack of previous spatial transcriptome data of oral mucosa tissue affects the reliability of this method; however, as the technology matures our limitations will be overcome particularly regarding resolution. Understanding the exact cellular and molecular mechanisms of oral mucosa cellular remodelling processes in disease in their spatial context will be key to improve our current understanding of oral mucosa physiology. In contrast to single-cell RNA sequencing methods, we are not treating or digesting the tissue with enzymes or extracting cells from their local environment, therefore the impact on gene expression is substantially inferior compared to single-cell RNA sequencing. Because of this key difference, we expect differences between single-cell RNA sequencing and spatial data, which can preclude successful data integration. We were not successful in mapping all fibroblasts using one strategy (anchor-based integration) because this integration is performed on low resolution Visium datasets which is unable to uncover fine cell subtypes, such as fibroblasts. When we performed integration using a higher spatial resolution method, we could map these cells.

In our initial single-cell RNA sequencing datasets, some gingiva cells were indeed missing due to technical limitations; for example, neutrophils were not captured given their fragile nature and low RNA content. With the spatial data, we could detect these and other immune cell types that were originally undetected. In conclusion, for a robust and unbiased molecular characterisation of human oral mucosa, spatial transcriptome data is essential.

2. Although the authors did the identification of the captured tissues, the results seem to require more analysis. Take Figure 5A as an example, there is a clear overlap between endothelial cells and basal cells. In addition, it is suggested that the authors indicate the specific location of the 10 clusters of cells in Figures 1D and 2C.

Thank you for your comment. Endothelial cells in Figure 5A have a predominantly subepithelial location as shown; however, these also localise in interpapillary regions which can be confounded with basal areas given the current resolution. We highlight that these analyses are not single-cell resolution. We applied a deconvolution method to increase the original spatial data resolution (55 µm), but it is still not true single-cell resolution.

In Figure 1D and 2C we are not showing clusters of cells, but spatial/anatomical cluster regions; for example, epithelial and stromal regions. These regions contain, especially stromal areas, information of multiple cell types. We can map epithelial regions as these are generally well defined (Figure 2F), but validating stromal regions becomes more difficult. To address this, we mapped individual cell types (Figures 5 and 6) and focused on locating and validating our cell type of interest (Fibroblast 5).

3. Healthy gingival tissues were taken from seven individuals. It would be helpful to know whether there was significant individual variation in cell clusters among these samples. Furthermore, regarding gingival tissue collected from patients with periodontitis, were these tissue samples collected from similar anatomical regions/locations?

Thank you for raising the importance of individual variability. We did not detect differences in the anatomical location of canonical structural cell types (e.g., basal keratinocytes, suprabasal keratinocytes); however, stromal cell types with a less defined positional identity or without a fixed tissue position (e.g., migrating immune cells) differed across patient samples as expected. In common across disease samples, we found that there was generally a highly immunogenic area where lymphoid and myeloid cells are highly expressed (stromal reticular regions). We have now added gene lists and hierarchical clustering analyses for each patient sample. These samples were all collected from first and second molars, except for patient 41 where tissues were collected from a second lower premolar.

Spatial clusters (not cell type clusters) differed in number due to different sample sizes and interindividual structural variation; for example, patients 35, 41 and 7 have a considerably small tissue surface area, therefore the number of anatomical clusters analysed are reduced.

4. In healthy gingiva, the authors identified dominant interactions between suprabasal and basal epithelial cells. The authors should comment on the biological significance of these interactions.

We thank the reviewer for the valuable suggestion. The oral epithelium exhibits a remarkable epithelial turnover, therefore understanding basal and suprabasal cell interactions is essential to understand tissue physiology. Furthermore, it has been reported that in the skin, stem cell renewal is induced by the differentiation of neighbouring cells (Mesa *et al.,* 2018, *Cell Stem Cell*), thus understanding the physiological factors that drive stem cell self-renewal will expand our knowledge of epidermal homeostasis and regeneration. Our high-resolution spatial mapping provides a remarkable new opportunity to address these questions. In our communication analyses, we detected high probability of interactions between ADAM17-NOTCH1. The roles of Notch signalling in proliferation and differentiation of stem/progenitor cells have been actively studied in various tissues, including the developing lung and intestine (Chiba, 2006; Tsao *et al.,* 2009; van der Flier and Clevers, 2009; Rock *et al.,* 2011). It is therefore possible that Notch signalling regulates both oral progenitor proliferation and differentiation. We have now added a comment in the Discussion section (Line 689).

5. Are those rare pro-inflammatory fibroblasts close to junctional epithelium?

Yes, from the patient samples where we can have a clear tissue orientation this population localises next to the junctional epithelium as shown in Figure 7 and Figure 6 – supplement 1. We have now added better image annotations for landmark structures (Figure 2 A,B).

6. The organization of the cell populations under healthy conditions and periodontitis is very different based on the data shown in Figure 1D and Figure 2C. The authors should compare these populations more clearly and explain the reason behind this.

Thank you very much for your suggestion. The clusters shown on Figure 1D and Figure 2C correspond to anatomical/spatial clusters, and not cell types. We agree that in an ideal experimental set up, patient samples would be collected with the exact same sampling; however, because each patient has distinct treatment needs, clinical samples will consequently have structural variation depending on the excision performed. The differences between 1D and 2C are essentially because in 2C the excision extended from buccal to interproximal and therefore we could profile oral, sulcular and junctional epithelium, whereas in 1D this distinction is not present. However, when we compare the same histomorphological regions (e.g., stromal oral subepithelial, 1G-2G), we observe similar features, such as terms enriched for ‘epidermis development’, suggesting that the function of specific anatomical landmarks are conserved across patients.

7. There are ten cell populations shown in Figure 1F and Figure 2D but the interactions highlighted in Figure 4 only have six groups. Could the authors explain this?

Thank you very much for highlighting this, and we should have made it clearer. The algorithm used for the communications analyses requires its own clustering step, and therefore the outcome is different from our high-resolution clustering analysis. We agree this is a major limitation and using a computational method that encompasses both analyses would provide more data cohesion. Despite the inferior clustering resolution of the ligand-receptor computational method, we could still detect critical regions of interest, such as basal-subepithelial and basal-suprabasal that provided novel insights into key signalling pathways *in situ.*

8. The heatmap including the differential gene expression profile needs to be shown for the fibroblasts annotated in Figure 6, in addition to the Dot plot. Moreover, the distribution of Fb1-Fb4 is quite similar under diseased conditions. What is the reason behind this?

Thank you for the suggestion; a heatmap for all oral mucosa fibroblasts and DGE list can be found in Caetano *et al.,* 2022, therefore we did not include it in this publication – we have added a reference. We don’t know the answer to why in disease there is less spatial segregation between the different fibroblasts. We can only hypothesise that since these cells acquire a pro-inflammatory phenotype perhaps, they become functionally impaired in disease with loss of homeostatic positional information. Another reason could be derived from the fact that these subtypes show small transcriptional differences even in health, therefore when in a pro-inflammatory environment, these differences become less obvious as they become dysregulated. Nevertheless, we still observe some differences in disease; for example, Fb2 can still be seen around the subepithelial region, and Fb3 does not seem to localise in the most immunogenic region.

Reviewer #1 (Recommendations for the authors):1. The results of "Molecular spatial reconstruction of the human oral mucosa in health" (Figure 1) are suggested to be validated using FISH or IHC. In addition, the authors need to describe the sample size of the study more clearly. Figure 1A shows a sample size of 7, but the main text describes 5 healthy samples, and 4 periodontitis samples are included.

We thank the reviewer for this important comment. We used a total of 7 biological replicates plus 2 technical replicates (Patients 43 and 45; Figure 1 —figure supplement 2) in order to validate our method. In short, we used 3 healthy biological replicates and 2 healthy technical replicates, and 4 disease biological replicates, totalling 9 10X Visium samples. We have now made this distinction clearer in the main text (Lines 123, 124). In order to validate all spatial regions using FISH or IHC (specifically stromal regions) would require a significant amount of probes/antibodies analysed simultaneously as one spatial region is defined by multiple cell types; however, we did validate known spatial regions such as suprabasal and basal epithelial regions, and we also validated our main finding showing that fibroblast 5 co-localises with other immune cell types in Figure 7.

2. Figure 2E shows the markers for each tissue compartment, and which of these 41 markers need to be tagged as pro-inflammatory genes to corroborate the authors' conclusion that "we also observed several disease-related changes in the stromal compartment. For example, the overall upregulation of pro-inflammatory genes."

Thank you – we agree there was not enough evidence to support these statements. We have now provided heatmaps of the top differential expressed genes for each region (Figure 1 – supplementary 1). To identify general tissue differences during disease remodelling, we compared the samples of distinct histomorphological regions and individuals at the molecular and compositional level. We observed common regions that we termed ‘immune-enriched’. Hierarchical clustering of the spatial transcriptomics clustering supports this finding; however, since the profile of each spatial transcriptomic dataset combines information of multiple cell types, we also assessed differences in cell composition determined by both spatial and single-cell RNA sequencing analyses (Figure 1—figure supplement 2; Figure 5—figure supplement 2). These immune-enriched regions in disease samples showed a larger proportion of lymphoid and myeloid cells (Figure 5; Figure 5—figure supplement 2, patient 35, patient 7), reflecting the expected role that immune cell interactions have in mucosal inflammation. In summary, these data enabled us to interindividual spatial differences, and to gain molecular insights of disease-specific spatial tissue remodelling.

3. The authors cited a reference (Nibali et al., 2019) to analyze the transcriptional defects that might drive the process of periodontitis, but no specific periodontitis-associated genes were given in this article. In addition, the "21 known disease gene" used in the article needs to be shown in the supplemental table. Since it is not known which 21 known disease genes were examined by the authors, it is not possible to determine "In summary, our analysis suggests that dysfunctional junctional epithelium might be a major driver of periodontal disease pathogenesis."

Thank you for your comment – we agree that we should have provided the full list of the periodontitis risk genes analysed. We have included a new reference to our previous GWAS study whereby it’s clear which 21 genes were analysed.

4. In Figure 2E, the image that validates the spatial transcriptome results, the results of KTR76 do not seem to match

Thank you for your comment; we agree that the immunostaining should have been from the same patient analysed in the spatial data shown on the left (Patient 30). However, both patient data show a clear matched suprabasal expression.

5. The authors concluded that "KRT13 and KRT4 being restricted to the sulcular epithelium and KRT13 specific to the junctional epithelium", but the results showed that KRT13 was not specifically distributed in the epithelium. The same was observed for SOX9.

Thank you for your comment; we agree that it is difficult to determine anatomical border zones with the data shown. We provided an H&E staining in Figure 2 for clarity, where we can superimpose molecular and morphological data. We agree that KRT13 also shows marginal expression in the stromal regions, and we attributed this to the current low resolution of this method; however, we do see a higher expression in the sulcular region, and this is also supported by the top differentially expressed genes for this spatial region (now in supplementary data). *SOX9* shows a more restricted expression in the junctional epithelium (Figure 3A). We believe this is particularly an important finding since details of the molecular networks controlling human oral progenitor self-renewal remain unknown. Future studies, addressing the role of *SOX9* in the junctional epithelium will provide essential insights into oral tissue repair and remodelling.

6. There are several errors in the Figures in the section "Effect of tissue microenvironments on epithelial identity and dynamics". For example, Figure 3B does not provide the results of FOXP2 and NEFM expression and Figure 2G has no results related to the signaling pathway.

Thank you very much – we have now corrected this and added FOXP2 and NEFM expression in Figure 3B. Do the reviewers mean Figure 1F instead of 2G? We added from line 299 a better explanation of these results by describing only Figure 3B in this section and not 1F.

7. Microenvironments are not well characterized by only looking at gene expression levels in the section "Effect of tissue microenvironments on epithelial identity and dynamics".

Thank you for your comment – we have now changed the title of this section to ‘Oral epithelia regional heterogeneity’.

8. According to the legend of Figure 4A, the abundant interactions are depicted in red. However, the results labeled in red in the healthy samples are not epithelial and basal epithelial layers which are not consistent with the authors' description that "in health we observed a major interaction between the epithelial and basal epithelial layers."

Thank you for your comment; in figure 4A (health) enriched interactions are predominantly epithelial, suprabasal-basal and epithelial immune2-epithelial immune interactions. We have relabelled the group ‘suprabasal’ to ‘suprabasal epithelial’ to reduce ambiguity, as well as ‘basal’ to ‘basal epithelial’.

9. Is there any other explanation for "we observed a substantial increase in endothelial cell expression and distribution in the oral subepithelial and reticular stroma, which contradicts our scRNA-seq analysis alone emphasizing the need to complement single-cell analyses with other data modalities."?

Thank you for pointing this interesting finding – in our reference single-cell dataset, endothelial cells were found reduced in disease, however, in our spatial data the opposite is observed. It is likely because endothelial cells may be impacted by the enzymatic treatment needed to digest the tissue into a single cell suspension. Because spatial data does not suffer from this limitation, we can better capture an accurate *in vivo* tissue profile. The same happened for neutrophils whereby these could not be detected in the original single cell dataset.

10. The markers of cells in Figures 5A and B are critical, but the authors do not provide this information.

Thank you very much – we strongly agree with this suggestion. We combined expression of marker genes derived from our single-cell transcriptome analysis (Caetano *et al.,* 2021), but also from available oral mucosa single cell datasets (Williams *et al.,* 2021, Huang *et al.,* 2021). We have now included in the text (line 367) these references.

11. The raw spatial sequencing data obtained from patients used in this study is deposited under GSE206621. Are the results of the healthy control tissue also in this dataset?

Yes, all patient samples used (healthy and disease) for spatial transcriptome analyses are deposited under GSE206621.

Reviewer #2 (Recommendations for the authors):There are some specific questions that will need to be addressed before this manuscript is ready for publication.1. Healthy gingival tissues were taken from seven individuals. It would be helpful to know whether there was significant individual variation in cell clusters among these samples. Furthermore, regarding gingival tissue collected from patients with periodontitis, were these tissue samples collected from similar anatomical regions/locations?

Thank you for raising the importance of individual variability. We did not detect differences in the anatomical location of canonical structural cell types (e.g., basal keratinocytes, suprabasal keratinocytes); however, cell types with less defined positional identity or without a fixed tissue position (e.g. migrating immune cells) differed across patient samples as expected. We found that in disease samples, there was generally a highly immunogenic area where various immune cells are highly expressed (stromal reticular regions). These samples were all collected from first and second molars, with the exception of patient 41 where tissues were collected from a second lower premolar.

Spatial clusters (not single cell clusters) differed in number due to different sample sizes; for example, patients 35, 41 and 7 have a considerably small tissue surface area, therefore the number of anatomical clusters analysed are reduced.

2. In healthy gingiva, the authors identified dominant interactions between suprabasal and basal epithelial cells. The authors should comment on the biological significance of these interactions.

We thank the reviewer for the highly valuable suggestion. The oral epithelium exhibits a remarkable epithelial turnover, therefore understanding basal and suprabasal cell interactions is essential to understand tissue physiology. Furthermore, it has been reported that in the skin, stem cell renewal is induced by the differentiation of neighbouring cells (Mesa *et al.,* 2018, *Cell Stem Cell*), thus understanding the physiological factors that drive stem cell self-renewal will expand our knowledge of epidermal homeostasis and regeneration. Our high-resolution spatial mapping provides a remarkable new opportunity to address these questions. In our communication analyses, we detected high probability of interactions between ADAM17-NOTCH1. The roles of Notch signalling in proliferation and differentiation of stem/progenitor cells have been actively studied in various tissues, including the developing lung and intestine (Chiba, 2006; Tsao *et al.,* 2009; van der Flier and Clevers, 2009; Rock *et al.,* 2011). It is therefore possible that Notch signalling regulates both oral progenitor proliferation and differentiation. We have now added a comment in the Discussion section (Line 689), and we look forward to investigate this interesting finding in future.

3. Are those rare pro-inflammatory fibroblasts close to junctional epithelium?

Yes, from the patient samples where we can have a clear tissue orientation this population localises next to the junctional epithelium as shown in Figure 7 and Figure 6 – supplement 1.

4. The organization of the cell populations under healthy conditions and periodontitis is very different based on the data shown in Figure 1D and Figure 2C. The authors should compare these populations more clearly and explain the reason behind this.

Thank you very much for your suggestion. The clusters shown on Figure 1D and Figure 2C correspond to anatomical/spatial clusters, and not cell types. We agree that in an ideal experimental set up, patient samples would be collected with the exact same sampling; however, because each patient has distinct treatment needs, clinical samples will consequently have structural variation depending on the excision performed. The differences between 1D and 2C are essentially because in 2C the excision extended from buccal to interproximal and therefore we could profile oral, sulcular and junctional epithelium, whereas in 1D this distinction is not present. However, when we compare the same identified regions (e.g. stromal oral subepithelial, 1G-2G), we observe similar features, such as ‘epidermis development’, suggesting that the function of specific anatomical regions are conserved across patients.

5. There are ten cell populations shown in Figure 1F and Figure 2D but the interactions highlighted in Figure 4 only have six groups. Could the authors explain this?

Thank you very much for highlighting this, and we should have made it clearer. The algorithm used for the communications analyses requires its own clustering step, and therefore the outcome is different from our high-resolution clustering analysis. We agree this is a major limitation and using a computational method that encompasses both analyses would provide more data cohesion. Despite the inferior clustering resolution of the ligand-receptor computational method, we could still detect critical regions of interest, such as basal-subepithelial and basal-suprabasal that provided novel insights into key signalling pathways *in situ.*

6. The heatmap including the differential gene expression profile needs to be shown for the fibroblasts annotated in Figure 6, in addition to the Dotplot. Moreover, the distribution of Fb1-Fb4 is quite similar under diseased conditions. What is the reason behind this?

Thank you for the suggestion; a heatmap for all oral mucosa fibroblasts and DGE list can be found in Caetano *et al.,* 2022, therefore we did not include it in this publication – we have added a reference. We don’t know the answer to why in disease there is less spatial segregation between the different fibroblasts. We can only hypothesise that since these cells acquire a pro-inflammatory phenotype perhaps they become functionally impaired in disease with loss of homeostatic positional information. Another reason could be derived from the fact that these subtypes show small transcriptional differences even in health, therefore when in a pro-inflammatory environment, these differences become less obvious as they become dysregulated. Nevertheless, we still observe some differences in disease; for example, Fb2 still can be seen around the subepithelial region, and Fb3 does not seem to localise in the most immunogenic region.